# *Neisseria gonorrhoeae* LIN codes provide a robust, multi-resolution lineage nomenclature

**Anastasia Unitt[1], Made A Krisna[2], Kasia M Parfitt[2], Keith A Jolley[2], Martin CJ Maiden[2], Odile B Harrison[1]***

[1]Nuffield Department of Population Health, University of Oxford, Oxford, United Kingdom; [2]Department of Biology, University of Oxford, Oxford, United Kingdom

## eLife Assessment

This **important** study introduces the Life Identification Number (LIN) coding system as a powerful and versatile approach for classifying Neisseria gonorrhoeae lineages. The authors show that LIN codes capture both previously defined lineages and their relationships in a way that aligns with the species' phylogenetic structure. The **compelling** evidence presented, together with its integration into the PubMLST platform, underscores its strong potential to enhance epidemiological surveillance and advance our understanding of gonococcal population biology.

**\*For correspondence:**
odile.harrison@ndph.ox.ac.uk

**Abstract** Investigation of the bacterial pathogen *Neisseria gonorrhoeae* is complicated by extensive horizontal gene transfer: a process which disrupts phylogenetic signals and impedes our understanding of population structure. The ability to consistently identify *N. gonorrhoeae* lineages is important for surveillance of this increasingly antimicrobial resistant organism, facilitating efficient communication regarding its epidemiology; however, conventional typing systems fail to reflect *N. gonorrhoeae* strain taxonomy in a reliable and stable manner. Here, a *N. gonorrhoeae* genomic lineage nomenclature, based on the barcoding system of Life Identification Number (LIN) codes, was developed using a refined 1430 core gene MLST (cgMLST). This hierarchical LIN code nomenclature conveys lineage information at multiple levels of resolution within one code, enabling it to provide immediate context to an isolate's ancestry, and to relate to familiar, previously used typing schemes such as Ng cgMLST v1, 7-locus MLST, or NG-STAR clonal complex (CC). Clustering with LIN codes accurately reflects gonococcal diversity and population structure, providing insight into associations between genotype and phenotype for traits such as antibiotic resistance. These codes are automatically assigned and publicly accessible via the https://pubmlst.org/organisms/neisseria-spp database.

## Introduction

Taxonomic classification is necessary to unravel evolutionary relationships, while also enabling the establishment of nomenclatures that facilitate effective communication of where an organism falls on a phylogeny (*Maiden et al., 2013*). This is particularly important for lineages within bacterial species, which are often distinct in clinically relevant phenotypes including antimicrobial resistance (AMR) and virulence (*Maiden et al., 2013*). The ability to identify these lineages consistently via a stable nomenclature system is essential to epidemiological surveillance (*Maiden et al., 2013*).

In *Neisseria gonorrhoeae*, the identification of AMR-associated lineages is particularly relevant. The gonococcus is a multi-drug resistant pathogen included in the WHO priority list for AMR (*WHO, 2018*; *Unemo and Nicholas, 2012*). Globally, gonorrhoea causes a high burden of disease with an

estimated 86.9 million new cases annually, which, if not successfully treated, can cause sequelae such as infertility and pelvic inflammatory disease (*Unemo and Nicholas, 2012*; *Rowley et al., 2019*). Facing the dual challenges of AMR surveillance and gonococcal vaccine development (*Lyu et al., 2024*), accurate characterisation of gonococcal population structure and consistent identification of key lineages is of utmost importance (*Gottlieb et al., 2019*; *Harrison et al., 2020*).

A variety of approaches has been applied to *N. gonorrhoeae* lineage taxonomy, which can cause confusion and miscommunication. Underlying these disparate nomenclatures is the complex population structure of the gonococcus; it is a highly recombinogenic organism, carrying out extensive horizontal gene transfer (HGT) between gonococci, and more rarely with other species (*O'Rourke and Stevens, 1993*; *Unitt et al., 2024*; *Manoharan-Basil et al., 2022*; *Bennett et al., 2014*). Polymorphisms are continually reassorted through this process, leading to low levels of linkage disequilibrium and disrupting clonal population structure (*O'Rourke and Stevens, 1993*; *Hanage, 2016*). Consequently, *N. gonorrhoeae* has been described as a 'sexual clone', reflecting the strong impact of HGT and the therefore weak clonal signal present in gonococcal genomes (*Harrison et al., 2020*; *Harrison and Maiden, 2021*). This low signal means that many molecular typing methods that are effective at capturing lineages in relatives such as *N. meningitidis* are not as reliable in *N. gonorrhoeae* (*Harrison et al., 2020*). The result has been a proliferation of differing approaches and nomenclatures seeking to capture *N. gonorrhoeae* taxonomy, particularly since 2010 (*Harrison et al., 2016*; *Kwong et al., 2016*; *Golparian et al., 2021*).

Most of these methods are Multi-Locus Sequence Typing (MLST) based. MLST is a widely used molecular approach for characterising bacterial isolates through the combination of alleles across a set of genes (an 'allelic profile'; *Maiden et al., 1998*). Each unique combination of alleles across the profile represents a sequence type (ST), which can be used, where they correspond with clonal inheritance, to characterise lineages (*Maiden et al., 2013*). Isolates can be further grouped based on these STs, using methods such as goeBURST (*Francisco et al., 2009*) or single linkage clustering (*Everitt et al., 2011*).

MLST is a powerful approach that has had widespread applications in the analysis of bacterial lineages (*Maiden et al., 2013*); however, conventional MLST approaches such as 7-locus MLST use only a small number of loci, which can make this system more vulnerable to disruptive HGT. In bacteria such as *N. gonorrhoeae*, HGT is extensive enough that it affects the housekeeping genes used in 7-locus MLST (*Harrison et al., 2020*). This makes 7-locus MLST a sub-optimal tool when applied in *N. gonorrhoeae* typing, as two isolates with the same ST may not actually be closely related, having come by the same ST by HGT rather than clonal inheritance (*Harrison et al., 2020*). This process affects several gonococcal typing systems, including NG-MAST (*Harrison et al., 2020*; *Kwong et al., 2016*) and NG-STAR (*Harrison et al., 2020*; *Golparian et al., 2021*).

Typing systems using larger numbers of loci can address this problem. For example, core gene MLST (cgMLST) is an extension of the MLST approach that uses a profile of hundreds of core genes to gain higher resolution, while also diluting the taxonomy-jumbling effects of HGT of some of those loci (*Maiden et al., 2013*; *Harrison et al., 2020*; *Maiden and Harrison, 2016*; *Palma et al., 2024*). In *N. gonorrhoeae* cgMLST v1, genes were defined as core if they were present in >95% of genomes, resulting in a core gene list (scheme) of >1600 core genes (*Harrison et al., 2020*). If two isolates were identical across this scheme, allowing for up to 50 loci to be unannotated, they would belong to the same core genome sequence type (cgST; *Harrison et al., 2020*). Single-linkage clustering was then applied to define core genome 'groups' based on various threshold levels of allelic differences within and between groups (*Harrison et al., 2020*). This clustering was necessary as the large number of loci used in cgMLST results in large numbers of unique cgSTs.

This method provided a high-resolution classification of *N. gonorrhoeae* lineage taxonomy, while being less affected by HGT than 7-locus MLST, NG-STAR or NG-MAST (*Harrison et al., 2020*). However, single-linkage clustering is susceptible to group fusion when intermediate isolates are found that bridge the identity thresholds between linkage groups (*Palma et al., 2024*; *Hennart et al., 2022*). HGT can exacerbate this issue, homogenising sequences across lineages by spreading polymorphisms throughout the population. As a result, the Ng cgMLST v1 core genome group nomenclature was not stable and changed over time as more *N. gonorrhoeae* isolates were sampled, decreasing the resolution provided and potentially leading to confusion as the nomenclature shifts. Therefore, there is still a need for a *N. gonorrhoeae* lineage nomenclature that provides

similarly effective discrimination between gonococcal lineages, while also remaining consistent over time.

Here, a definitive nomenclature for *N. gonorrhoeae* is proposed, using LIN codes. This isolate barcoding system has previously been applied to *Klebsiella pneumoniae* (*Hennart et al., 2022*) and *Streptococcus pneumoniae* (*van Rensburg et al., 2024*), where LIN codes were shown to provide a high resolution nomenclature, enabling accurate insight into the population structure of these organisms. LIN code combines the HGT-diluting effects of a cgMLST scheme with the stability of a numeric barcode (*Palma et al., 2024*; *Hennart et al., 2022*). This stability is derived from the fixed nature of these barcodes for each isolate, along with the flexibility of a multi-position numeric code which can increase infinitely as new variants are sequenced (*Palma et al., 2024*; *Hennart et al., 2022*; *Vinatzer et al., 2017*). The *N. gonorrhoeae* LIN code developed here was implemented in PubMLST, a freely accessible bacterial genomics database (RRID:SCR_012955) (*Jolley et al., 2018*). LIN codes are automatically assigned to all WGS uploaded to PubMLST, making this taxonomy easily applicable to old and new datasets, and encouraging the widespread use of this reliable nomenclature.

## Methods

### Representative isolate collections

Data were extracted from the PubMLST database (https://pubmlst.org/organisms/neisseria-spp), a publicly accessible bacterial genomics database (*Jolley et al., 2018*). At the time of writing, sequence records from over 28,000 gonococci were available in the database. To facilitate efficient analyses, representative sub-datasets were generated.

Dataset 1 consisted of a representative collection of 896 isolates belonging to a range of core genome groups at the 300 allelic mismatch threshold (*Figure 1*). This was assembled by including all isolates belonging to core genome groups represented by six or fewer isolates, in combination with a random selection of six isolates from each of the core genome groups represented by more than six isolates (*Supplementary file 1*). Randomisation was achieved in R, using the sample function (*R Development Core Team, 2018*). This smaller dataset was used for preliminary analysis, as it required less computational power.

Dataset 2 consisted of 3935 isolates (*Figure 1*), assembled by including all isolates belonging to core genome groups represented by 20 or fewer isolates (again at allelic mismatch threshold 300), and a random sample of 30% of all core genome groups represented by more than 20 isolates, capped at 350 isolates per core genome group (*Supplementary file 2*). This larger dataset was used for final analysis once techniques had been explored using dataset 1.

### Development of Ng cgMLST v2

Although a 1649 locus Ng cgMLST v1 scheme for *N. gonorrhoeae* has been published (*Harrison et al., 2020*), an updated, more readily auto-annotated cgMLST scheme was needed to ensure any WGS deposited in the PubMLST database can be assigned a LIN code with little to no manual curation (*Hennart et al., 2022*; *van Rensburg et al., 2024*). This was a consequence of Ng cgMLST v1 including a number of genes with alternate start codons or length variation (e.g. due to internal stop codons or frameshift mutations), complicating automatic allele annotation by PubMLST.

WGS data from dataset 1 were downloaded from PubMLST and annotated with Prokka using default parameters (RRID:SCR_014732) (*Seemann, 2014*). The resulting .gff output was further analysed with PIRATE (version 1.0.5) for pangenome analysis including the identification of core genes (RRID:SCR_017265; *Bayliss et al., 2019*). PIRATE was run using default parameters with the following options: -a (align all genes), -r (plot summaries using r), and –t 24 (24 threads). A preliminary threshold of 95% presence was chosen as 'core' rather than 100% to ensure a large enough gene list was identified. This also allowed for the possibility that some genes may be absent due to factors such as misassembly or mutation while still being 'core' in nature. Genes with duplication events as detected by PIRATE were excluded.

The new core list was assessed in dataset 2 to identify possible curation issues across a wider range of isolates. Following this analysis, all loci that did not meet a threshold of 98% presence in dataset 2 were excluded. This reduced Ng cgMLST v2 prioritised loci that annotate with very little human involvement via manual curation.

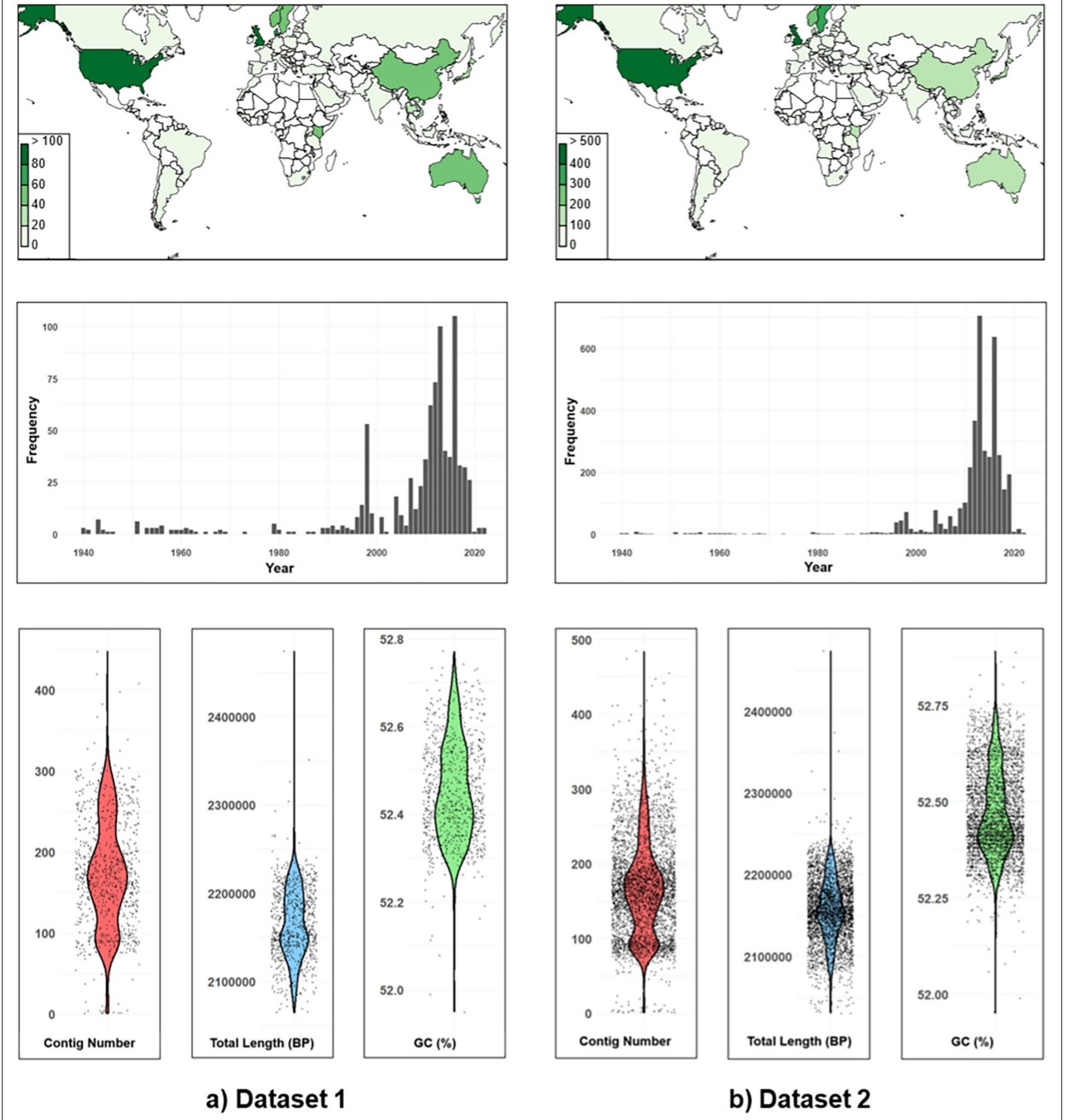

**Figure 1.** Characteristics of isolates within representative development datasets 1 and 2. This includes geographical distribution (top panels), frequency of isolates sampled over time (middle panels), and genome quality statistics (lower panels) including (i) contig number, (ii) total genome length and (iii) % GC content.

The PubMLST GrapeTree plugin was used to generate minimum spanning trees based on the Ng cgMLST v2 loci, annotated with core genome groups (based on Ng cgMLST v1), to ensure resolution was high enough with this new reduced scheme to explore established gonococcal lineages (*Zhou et al., 2018*). GrapeTree compares isolates across a user-defined allelic profile, clustering those that share the most alleles (*Zhou et al., 2018*).

## Analysis of population structure and threshold selection

Thresholds for LIN code were chosen based on an analysis of natural discontinuities in the gonococcal population structure, assessed by examining the distribution of pairwise allelic mismatches amongst

isolates from dataset 2, combined with testing in a local installation of LIN code (*Hennart, 2019*) and the application of clustering statistics including Rand index and silhouette score.

In order to extract pairwise allelic distances for these analyses, a distance matrix was generated comparing dataset 2 isolates across the loci included in Ng cgMLST v2. This was accomplished using the genome comparator plugin in PubMLST. This tool provides gene-by-gene pairwise analysis of isolates across a list of loci, generating a variety of outputs including a distance matrix, core gene list, and alignments (*Jolley et al., 2018*). Here, genome comparator was run with default settings. The distances in this matrix were stacked to form a dataframe with three columns (isolate A, isolate B, distance), allowing the distances to be plotted in a histogram using the ggplot2 package in R (RRID:SCR_014601) (*Wickham, 2016*).

Ridgeline plots were created based on these data to explore the number of allelic mismatches associated with clusters identified by pre-existing typing systems such as NG-STAR clonal complexes and Ng cgMLST v1 core genome groups. This was done by extracting the pairwise allelic distances associated with each instance of matching NG-STAR CC, rMLST-ST or matching Ng cgMLST v1 core genome group, from the distance matrix. Ridgeline plots were constructed in R using the packages ggplot2 and ggridges (RRID:SCR_024511; *Wickham, 2016*; *Wilke, 2024*).

Based on the observed breaks in population structure, various bin thresholds were tested using a local version of LIN code downloaded from https://gitlab.pasteur.fr/BEBP/LINcoding applied to dataset 2 (*Hennart, 2019*). The 3935 isolates yielded 3877 unique cgSTs for testing.

Clustering output from the local LIN code testing using various allelic mismatch thresholds was assessed using statistics, specifically the silhouette score and Rand index. The silhouette score indicates the cohesion of clustering at a particular threshold, varying from –1–1 with more positive values indicating higher cluster cohesiveness (*Rousseeuw, 1987*). The average silhouette score was calculated using MSTClust (v0.21b) (downloaded from https://gitlab.pasteur.fr/GIPhy/MSTclust) for pairwise distance thresholds from 0.01 to 0.7 (*Criscuolo, 2021*). Results were plotted using the R package ggplot2 (*Wickham, 2016*).

The Rand index is a statistical measure that indicates the level of similarity between two different partitions of the same dataset, with values approaching 1 indicating higher similarity (*Rand, 1971*). The Rand index was applied here to compare local LIN code clustering at various thresholds against Ng cgMLST v1 core genome group clustering at threshold 300 and 400. The index was calculated in R using the package fossil (*Vavrek, 2011*).

Finally, LIN codes generated using these thresholds on the representative dataset were visualised on a minimum spanning tree to explore what length of code prefix would be most suited to exploring population structure at various resolutions.

## Implementation and analyses

Once the refined Ng cgMLST v2 scheme was defined in PubMLST and isolates annotated with sequence types, the chosen thresholds were implemented within the database to generate LIN codes. A maximum of 25 missing loci in the Ng cgMLST v2 scheme was tolerated. If this quality threshold was not met, no cgST would be assigned to the isolate, and hence no LIN code. LIN codes were designated in PubMLST ordered based on a minimum spanning tree, using a default batch size of 10,000 isolates (*Palma et al., 2024*). Multilevel single-linkage clustering was used to classify isolates at each threshold within the code. The combination of these clusters with a fixed bar code facilitates LIN code's stability (*Hennart et al., 2022*; *van Rensburg et al., 2024*). Subsequently, isolate records that have a cgST but no LIN code assigned will be scanned once a week, and LIN codes will be automatically designated in batches, starting at 23:00 (UK local time) each Sunday. This includes records kept in private projects or embargoed, as well as public datasets.

Once LIN codes were designated within the database, PubMLST was further applied to investigate the distribution of LIN codes and to examine their association with other isolate typing systems such as 7-locus MLST. This was undertaken using the 'Field breakdown' and 'Combinations' analysis tools, alongside exporting datasets for manipulation.

A published dataset of 171 ceftriaxone-resistant gonococci from diverse phylogroups (*Fifer et al., 2015*) was selected for analysis, to validate LIN code's ability to reproduce complex phylogenies. Isolates from this dataset not already present in PubMLST were downloaded from the sequence read archive, and where necessary were assembled using SPAdes (version 3.15.4-GCC-12.3.0,

RRID:SCR_000131; *Prjibelski et al., 2020*). These isolates were then uploaded to PubMLST and associated under the appropriate publication record (*Fifer et al., 2015*). One isolate (H18-368) out of 171 no longer had sequence data available, and so could not be included in this analysis.

## Phylogenetic trees

Minimum spanning trees were drawn using the GrapeTree plugin in PubMLST (*Zhou et al., 2018*). Neighbour joining trees were constructed using the ITOL plugin within the PubMLST interface.

Nucleotide alignments for other trees were generated via genome comparator in PubMLST (*Jolley et al., 2018*), using the MUSCLE algorithm (RRID:SCR_011812), with settings to align all loci, not only variable loci. Maximum likelihood trees were constructed using RaxML (version 8.2.12-hybrid-avx2 gompi/2021b, RRID:SCR_006086; *Stamatakis, 2014*), and approximate maximum likelihood trees using FastTree (version 2.1.1, RRID:SCR_015501; *Price et al., 2010*). Trees were corrected for recombination using ClonalFrameML (version 1.13, RRID:SCR_016060; *Didelot and Wilson, 2015*) and edited using ITOL (*Letunic and Bork, 2021*).

Specific parameters and scripts for all programs have been documented at https://www.protocols.io/ (*Unitt, 2025*).

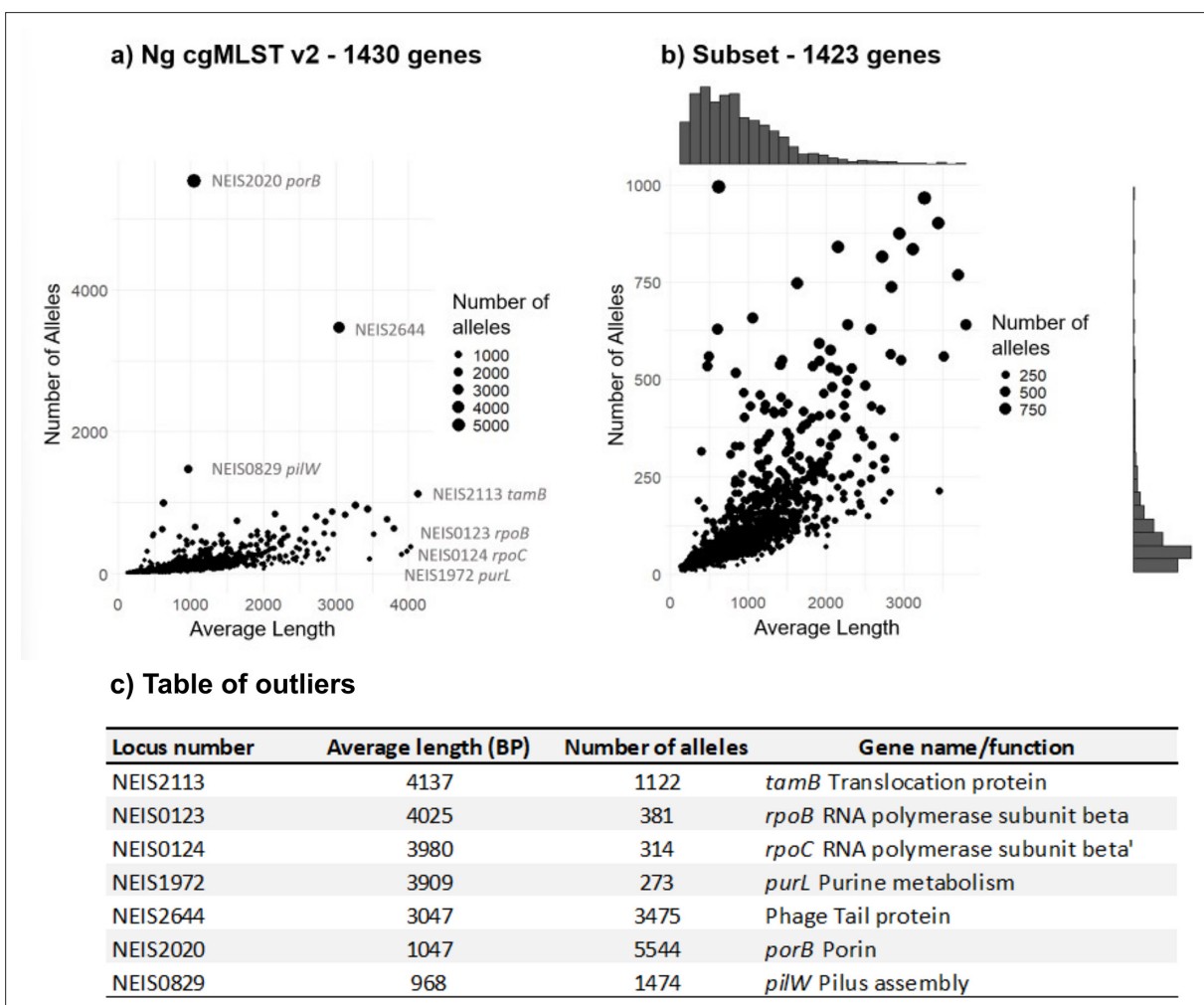

### c) Table of outliers

| Locus number | Average length (BP) | Number of alleles | Gene name/function |
| --- | --- | --- | --- |
| NEIS2113 | 4137 | 1122 | *tamB* Translocation protein |
| NEIS0123 | 4025 | 381 | *rpoB* RNA polymerase subunit beta |
| NEIS0124 | 3980 | 314 | *rpoC* RNA polymerase subunit beta' |
| NEIS1972 | 3909 | 273 | *purL* Purine metabolism |
| NEIS2644 | 3047 | 3475 | Phage Tail protein |
| NEIS2020 | 1047 | 5544 | *porB* Porin |
| NEIS0829 | 968 | 1474 | *pilW* Pilus assembly |

**Figure 2.** Allele number vs allele length for the 1430 genes in Ng cgMLST v2. (**a**) All 1430 genes included in Ng cgMLST v2 are plotted. (**b**) The four genes with the highest average length, and the four with the highest number of alleles, were excluded as outliers. The figure was compiled excluding these genes in order to allow a closer examination of the distribution of allele length vs number. One gene, NEIS2113, appeared in both lists. (**c**) Table summarising the average length in base pairs, number of alleles and gene name/function of the seven outlier loci.

## Results

### Core genome MLST version 2

The final Ng cgMLST v2 included 1430 loci. Of these, 362 were hypothetical genes of unknown function, 96 encoded transferases, 56 synthases, 45 transporters, 34 50 S ribosomal proteins, 29 transcriptional regulators and 21 30 S ribosomal proteins (*Supplementary file 3*). This represented a decrease of 219 loci compared to Ng cgMLST v1, with the largest categories of excluded loci being hypothetical (83 loci) or phage associated (18 loci; *Supplementary file 4*). Notable exclusions include *tbpA* and *tbpB*, essential iron acquisition proteins that are hypervariable and so not readily auto-annotated. In total, 1418 loci were shared between the two schemes, with 12 new to Ng cgMLST v2 (*Supplementary file 4*).

Of the 1430 loci included, 99% (1418/1430) had under 1000 alleles and an average length of less than 3000 base pairs (*Figure 2*). Exceptionally variable genes included NEIS2020 *porB* (5544 alleles), NEIS2644 an unnamed phage tail protein (3475 alleles), NEIS0829 *pilW* (1474 alleles), and NEIS2113 *tamB* (1122 alleles; *Figure 2c*).

It should be noted that while the new scheme is fundamentally still a core genome MLST scheme, it does not include all core genes and should not be used as a definitive core gene list. However, the strict inclusion criteria for Ng cgMLST v2 facilitated automated assignment of a cgST to a higher proportion of WGS data than its predecessor, while still providing a similar degree of resolution.

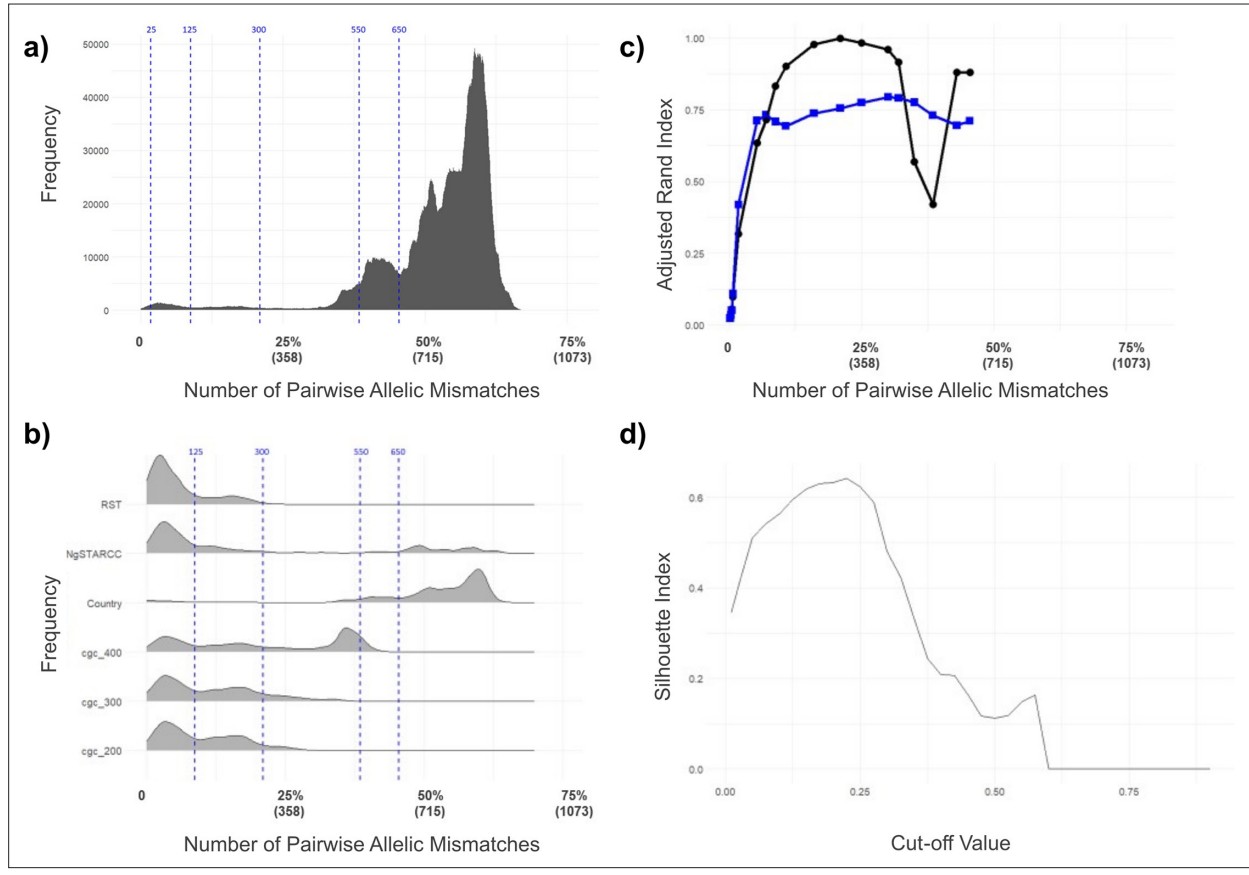

**Figure 3.** Plots used in the selection of allelic mismatch thresholds for the LIN code. (**a**) Histogram showing the frequency of pairwise allelic mismatches within dataset 2. A subset of the allelic mismatch thresholds applied in the gonococcal LIN code is shown (blue dashed lines) at 25 mismatches (1.75%), 125 (8.74%), 300 (20.98%), 550 (38.46%), and 650 (45.45%). (**b**) Ridgeline plots depicting the frequency of allelic mismatches amongst pairs of isolates that belong to the same category of different metrics from dataset 2. From top to bottom: Ribosomal MLST (RST), NG-STAR Clonal Complex (NgSTARCC), Country, Ng cgMLST v1 core genome group at threshold 400 (cgc_400), threshold 300 (cgc_300), and threshold 200 (cgc_200). (**c**) Plot of adjusted Rand index comparing LIN code clustering at various allelic mismatch thresholds to Ng cgMLST v1 core genome groups at threshold 300 (black dots) and NG-STAR CC (blue squares). Clustering was compared using dataset 2. (**d**) Plot of silhouette index (score) at various cutoff values, based on MSTclust analysis of 1430 core loci across 3935 representative *N. gonorrhoeae* isolates (dataset 2). Silhouette score peaked at 0.64 at a cutoff value of 0.225.

## Analysis of population structure and allelic mismatch thresholds

The number of allelic mismatches present in a pairwise comparison across the 3935 isolates in dataset 2 was assessed across the 1430 core genes in Ng cgMLST v2. When visualised as a histogram (*Figure 3a*), an uneven distribution of allelic mismatches was observed, exhibiting a highly varied incidence. The allelic mismatch modes indicate clusters of isolates that share the same proportion of alleles in common across the core genome scheme, reflecting natural breaks in the gonococcal population structure (*Hennart et al., 2022*).

The most frequently identified number of allelic mismatches between isolates was 840/1430 (59%). This modal mismatch is low compared to that seen in *K. pneumoniae* (within species mode 520/629 mismatches, 83%; *Hennart et al., 2022*). There were no instances of pairwise allelic mismatches exceeding 970/1430 loci (68%), meaning that all isolates shared a minimum of 460 alleles with one other isolate in the dataset. This means that compared to *K. pneumoniae*, *N. gonorrhoeae* isolates shared more of the same alleles across their core genome and were therefore less diverse. This may be due to the homogenising effect of widespread HGT.

When pairwise allelic mismatches were analysed in isolates belonging to the same grouping, such as Ng cgMLST v1 core genome groups or NG-STAR CCs, discontinuities in the population structure associated with these metrics became visible (*Figure 3b*). For isolates belonging to the same core genome group (at threshold 300), NG-STAR CC, or ribosomal ST, most pairs had under 125/1430 allelic mismatches between them (*Figure 3b*). This indicates that shared identity in approximately 1305/1430 (91%) Ng cgMLST v2 loci is required to belong to the same core genome group (at threshold 300), NG-STAR CC, or ribosomal ST. However, core genome groups at threshold 400 displayed a second peak at 510/1430 (36%) mismatches. This is a result of group fusion due to intermediate genotypes, which has led to divergent gonococci being included in the same group at this threshold. Isolates from the same country displayed a comparatively high average level of allelic mismatch, with a mode of 855/1430 (60%), indicative of limited geographical association of gonococcal lineages.

Rand index values when comparing LIN code bin thresholds to Ng cgMLST v1 core genome groups (at threshold 300) peaked at 1 for a 300-locus mismatch threshold (21%) (*Figure 3c*). Similarly, the silhouette score demonstrated a peak of 0.64 at cut-off 0.225 (*Figure 3d*).

Based on this evidence, allelic mismatch thresholds were chosen for each bin of the LIN code. Thresholds at 125 loci (8.74% mismatch) and 300 (20.98% mismatch) were chosen to correspond to discontinuities in population structure (*Figure 3a*), alongside a Rand index close to 1 demonstrating their relevance to core genome groups, and silhouette score indicating high cluster cohesiveness at these thresholds (*Figure 3c and d*). Higher thresholds at 550 loci (38.46% mismatch) and 650 (45.45%) captured superlineage divisions.

To provide the higher resolution necessary to identify transmission events or outbreak-related gonococcal variants, thresholds at lower levels of allelic mismatch were included. Together, this resulted in a set of 11 bins, corresponding to mismatch thresholds: 650, 550, 300, 125, 25, 10, 7, 5, 3, 1, and 0 (*Figure 4*). Minimum spanning tree analysis indicated that a prefix using the first three thresholds was ideal for exploring lineages (*Figure 5*) corresponding to Ng cgMLST v1 core genome groups at threshold 300.

## Implementation of LIN code

Across the >28,000 publicly available *N. gonorrhoeae* genomes stored in PubMLST, 25,912 had the minimum 1405/1430 core genes annotated, and therefore were assigned a LIN code. At the superlineage level (2 threshold prefix: 0_0) there were 118 clusters, at lineage (3 prefix: 0_0_0) 532 clusters, and at sublineage (4 prefix: 0_0_0_0) 1712 clusters. The 3-threshold prefix lineage definitions directly correlated with Ng cgMLST v1 core genome groups at threshold 300 (adjusted [adj.] Rand Index = 1). This was visualised using phylogenetic trees (*Figure 6*) and minimum spanning trees (*Figure 5*) which showed good congruence between LIN codes, SNP-based phylogeny, and other clustering methods. The lineages identified by LIN code demonstrated persistence over time, with lineage 0_2_1 isolates spanning 39 years from 1985 to 2024 (*Figure 5—figure supplement 1*).

*N. gonorrhoeae* LIN codes were accessed via an isolate's information page, from the 'allele designations/scheme fields' dropdown box within the https://pubmlst.org/organisms/neisseria-spp isolate database search form, or through the 'export dataset' link at the bottom of the page following an isolate search. They could also be exported as metadata in GrapeTree analyses.

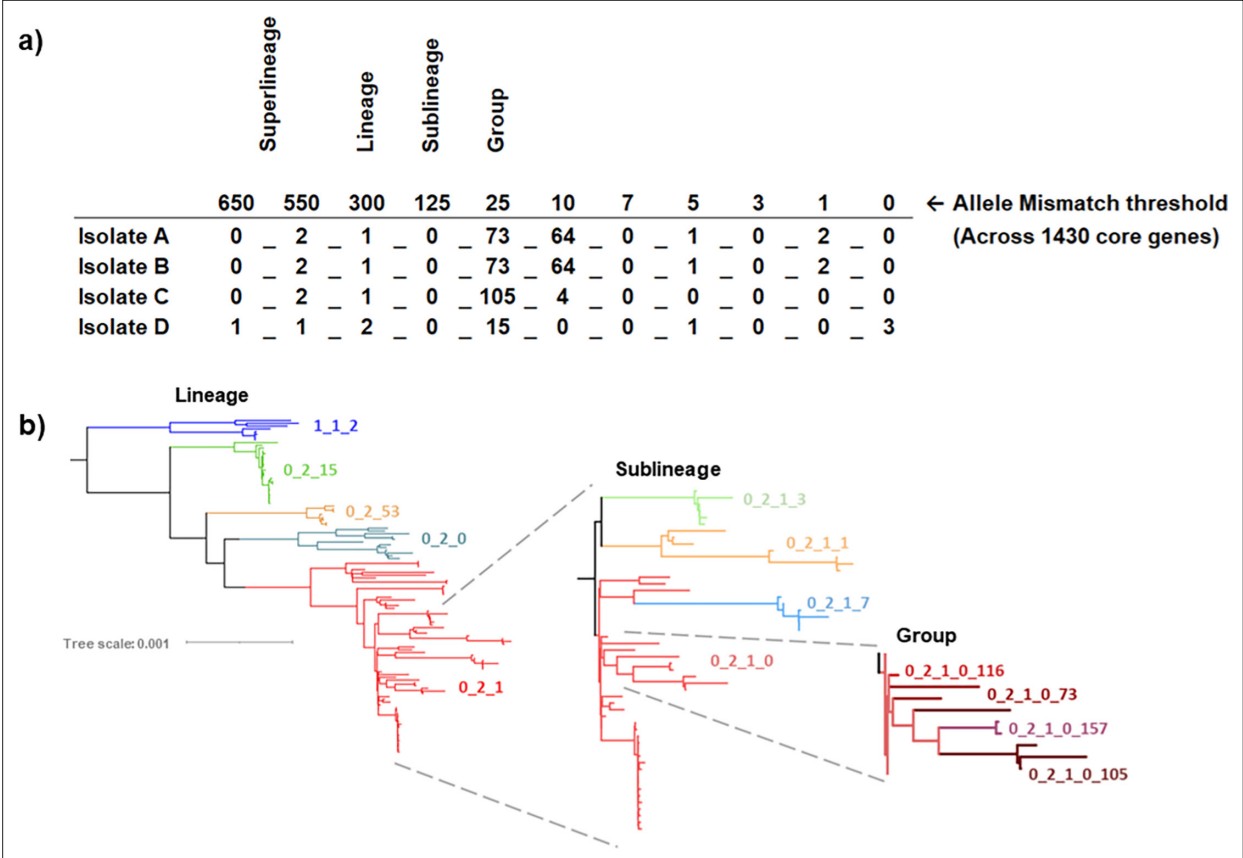

**Figure 4.** Illustration of the gonococcal LIN code nomenclature. (**a**) Each successive allelic mismatch threshold dictates clustering at a specific position within the code. This clustering is hierarchical, such that isolates sharing a larger proportion of the code (from the left across) are of higher genetic similarity. For example, Isolate A and B share a complete LIN code, meaning they have 0 allelic mismatches in their Ng cgMLST v2 loci. Isolate B and C share the first three digits of their LIN code; they belong to the same clusters at these thresholds, and therefore differ in less than 300 alleles out of the 1430 core genes in Ng cgMLST v2 i.e., they belong to the same LIN code "lineage". (**b**) Rooted Maximum likelihood tree demonstrating how LIN codes reflect phylogenetic relationships. The first tree shows a subset of LIN code lineages within superlineage 0_2, with lineage 1_1_2 as the outgroup. Moving to the right, the figure focuses in on lineage 0_2_1, showing the higher resolution provided by LIN sublineages, and then groups. (Figure inspired by Figure 3 in ***van Rensburg et al., 2024***; ***R Development Core Team, 2018***).

## Congruence with previous typing schemes

7-locus MLST-STs showed good association with LIN code lineages (0_0_0; adj. Rand index = 0.92) (***Figure 5***), although many STs included multiple lineages. For example, MLST-ST 1901 corresponded with lineages 0_2_0 (1932/2376, 81%) and 0_0_12 (443/2376, 19%), while MLST-ST 7363 was predominantly captured by 0_0_52 (796/1322, 60%) and 0_2_17 (355/1322, 27%; ***Supplementary file 5***). The fact that these LIN codes are from different superlineages (0_0 and 0_2) illustrates how these MLST-STs form polyphyletic groupings, combining isolates that are not closely related under the same ST.

NG-STAR clonal complexes (CCs) showed a stronger association with LIN code sublineages (0_0_0_0; adj. Rand index 0.96; ***Figure 5***). NG-STAR CC 90 belonged predominantly to sublineages 0_2_0_1 (1499/1562, 96%) and to a much lesser extent 0_0_12_2 (25/1562, 2%). NG-STAR CC 63 was represented mainly by 0_2_1_10 (1369/1437, 95%), followed by 1_1_22_9 (21/1437, 1%) and others (***Supplementary file 5***). Again, the diversity of these LIN codes at the superlineage level, and comparison with their location on the phylogeny (***Figures 5 and 6***), demonstrates how genetically divergent isolates can belong to the same CC.

NG-MAST-STs showed some correlation with LIN code groups (0_0_0_0_0; adj. Rand index 0.50). For example, NG-MAST-ST 2992 belonged mostly to group 0_2_1_0_73 (409/653, 63%) and to others such as 0_2_1_1_4 (72/653, 11%). NG-MAST-ST 1407 predominantly belonged to group 0_2_0_1_0 (498/787, 63%) and to 0_2_0_1_27 (77/787, 10%).

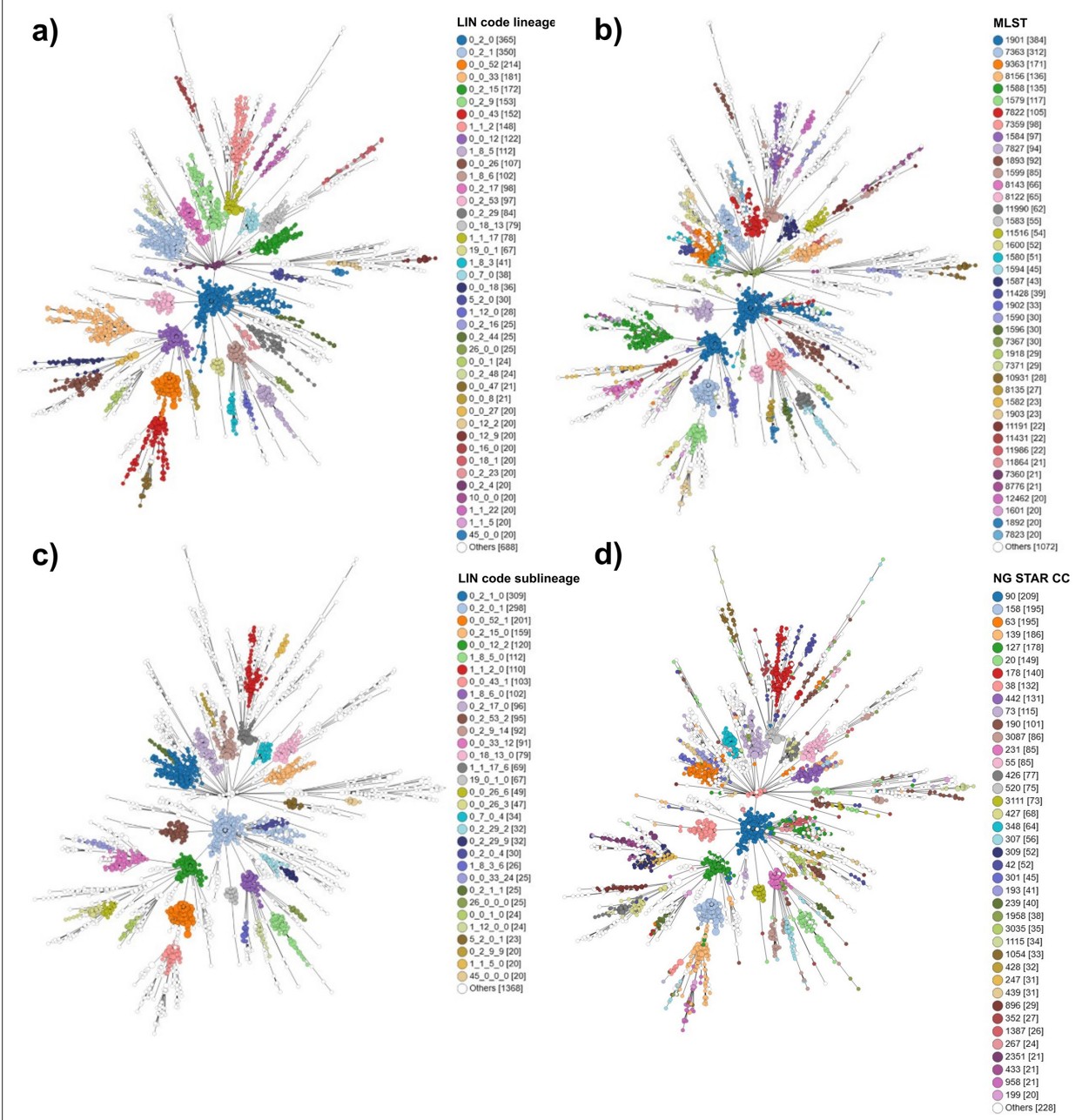

**Figure 5.** Minimum spanning tree showing clustering of 3935 isolates from dataset 2 based on Ng cgMLST v2. LIN code lineages (**a**) and 7-locus MLST (**b**) demonstrate similar levels of resolution for characterising clustering. LIN codes sublineages (**c**) provide higher resolution, similar to that provided by NG-STAR clonal complexes (**d**). Only categories including 20 or more isolates are coloured.

The online version of this article includes the following figure supplement(s) for figure 5:

**Figure supplement 1.** LIN lineage by year.

Resistance-associated lineage A, as described using hierarchical Bayesian analyses (BAPs; *Sánchez-Busó et al., 2019*), was previously shown to associate well with Ng cgMLST v1 core genome groups (*Harrison et al., 2020*). As LIN code lineages correspond to core genome groups at threshold 300 (*Figure 6*), the same is true of this nomenclature. Lineage 0_2_0, 0_2_1 and 0_2_17 correspond to core genome group 3, 16 and 8, which in turn correlate with BAPS 8, 7, and 9 respectively, all within the resistance associated lineage A (*Harrison et al., 2020*; *Sánchez-Busó et al., 2019*).

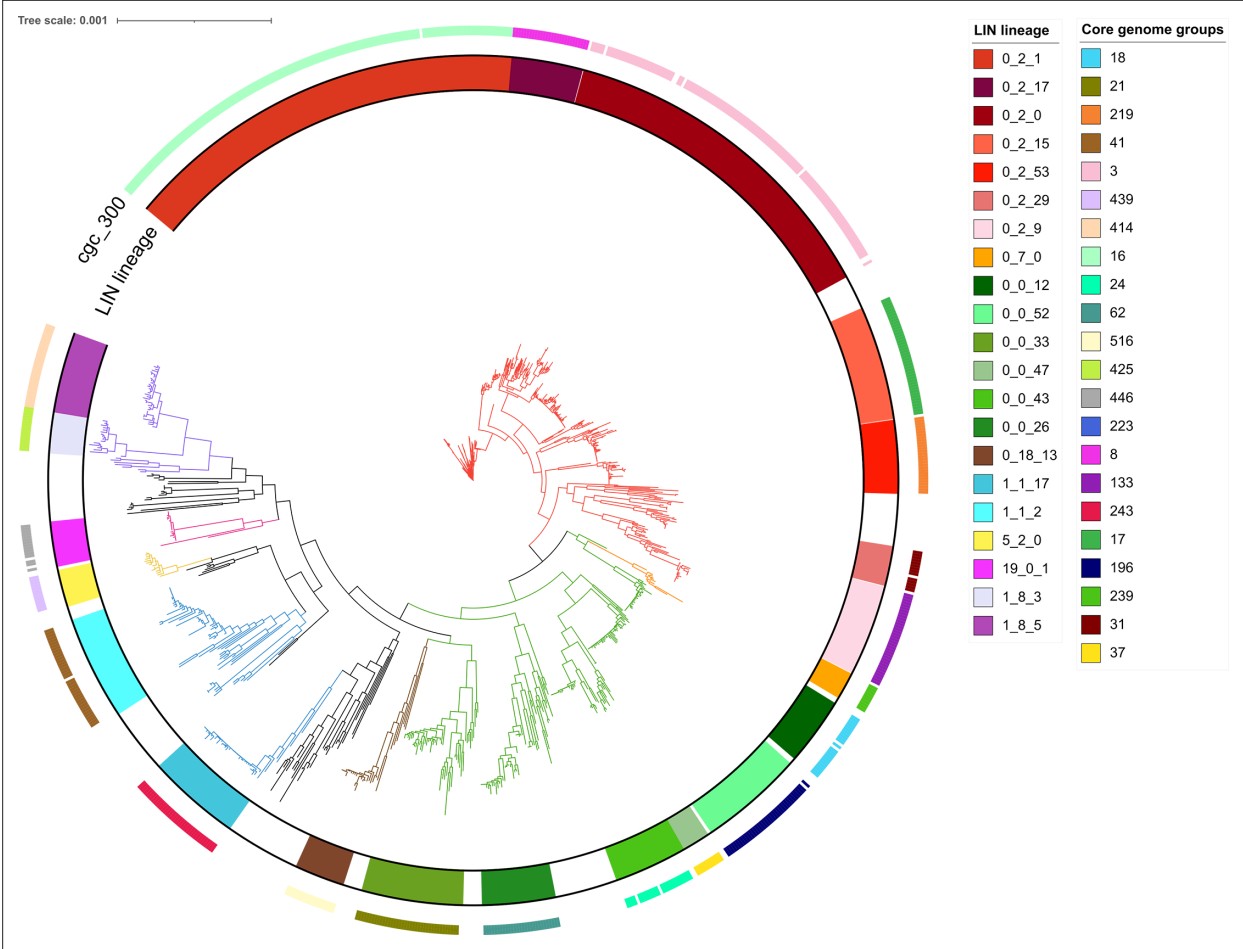

**Figure 6.** Unrooted FastTree of 1000 randomly chosen *N. gonorrhoeae* isolates with LIN codes assigned. Constructed using 1430 loci from Ng cgMLST v2. Branches are coloured by the 8 most frequent superlineages (0_2=red, 0_7=orange, 0_0=green, 0_18=brown, 1_1=blue, 5_2=yellow, 19_01=pink, and 1_8=purple.) The 21 highest frequency LIN lineages are labelled, represented by the inner bar, in colour ranges corresponding to their superlineage colour. LIN codes form monophyletic groupings, indicating that there is a good degree of congruence between the allelic profile clustering method used in cgMLST LIN code and nucleotide sequence alignment-based phylogeny. Core genome groups at threshold 300 (cgc_300) are represented by the outer coloured bar and show good concordance with LIN lineages, although fewer isolates were able to be annotated with core genome groups than LIN codes due to use of the larger and more poorly auto-annotated cgMLST v1. All isolates used in this tree had a LIN code assigned.

Mosaic *penA* type 34 alleles, an important antimicrobial resistance determinant, correlated with LIN code lineage; 89.43% of these alleles were found in isolates belonging to lineage 0_2_0 (***Supplementary file 6***).

## Example analysis

The gonococcal LIN code was applied in a reproduction of an analysis of 170 ceftriaxone-resistant gonococci initially performed by ***Fifer et al., 2015*** (***Supplementary file 7***). A maximum likelihood tree of these isolates was reconstructed and labelled using LIN codes (***Figure 7***), and demonstrated that LIN code lineages recaptured the diverse clades identified by phylogenetic methods, while also classifying several isolates that were not assigned a phylogroup in the original publication. Furthermore, 15 pairs or triplets of isolates sharing a full-length LIN code were identified within this dataset. Comparison with a random selection of isolates from across the PubMLST database (***Figure 6***) suggested an overrepresentation of 0_14_0 isolates in the ceftriaxone-resistant dataset. This dataset can be accessed by filtering by publication '***Fifer et al., 2015***' on the PubMLST *Neisseria* isolate search page.

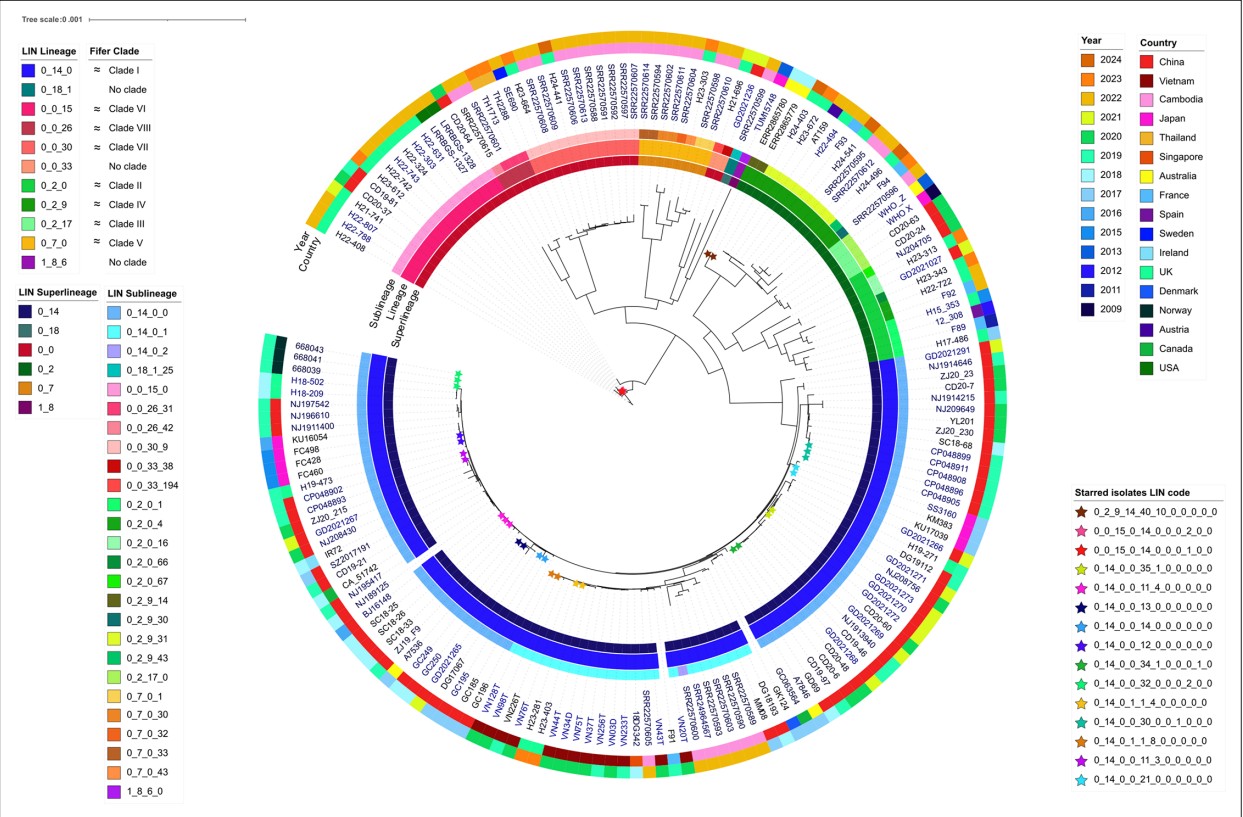

**Figure 7.** Maximum Likelihood tree of 170 Ceftriaxone-resistant isolates previously analysed in *Fifer et al., 2015*. Constructed using 1430 loci from Ng cgMLST v2. LIN code lineages were able to reproduce the clades identified in Fifer et al., while being readily accessible and providing additional detail about each clade in the form of superlineage and sublineage divisions. Groups of isolates that share the same full length LIN code are highlighted as coloured stars.

## Discussion

Accurate identification of bacterial lineages is necessary to examine links between bacterial population structure and characteristics such as AMR, and to enable surveillance of emergent or outbreak-associated variants (*Maiden et al., 2013*). However, in *N. gonorrhoeae,* widespread HGT reassorts DNA among isolates, disrupting clonal inheritance and consequently distorting phylogenetic trees (*Harrison et al., 2020*; *Turner and Feil, 2007*). This makes it difficult to accurately characterise gonococcal lineages, particularly when using conventional typing systems based on small numbers of genes, such as 7-locus MLST (*Harrison et al., 2020*; *Hanage, 2016*). Previously, the most reliable approach applied core genome MLST (cgMLST) to define core genome groups, using over 1000 genes to dilute the effects of HGT and facilitate high-resolution analyses (*Maiden et al., 2013*; *Harrison et al., 2020*; *de Korne-Elenbaas et al., 2022*). Unfortunately, this method suffers from cluster instability (*Hennart et al., 2022*; *Turner and Feil, 2007*). LIN code provides an alternative means to leverage the benefits of cgMLST within a stable classification system, which can accommodate new genotypes without the risk of disrupting established clustering (*Palma et al., 2024*; *Hennart et al., 2022*). Here, we developed a *N. gonorrhoeae* LIN code: an effective nomenclature for categorising, exploring, and discussing gonococcal population structure.

Evidence for the existence of distinct, persistent gonococcal lineages pre-dates whole genome sequencing, in the characterisation of auxotypes: related isolates that exhibit similar patterns of growth in media containing different combinations of key nutrients (*Catlin, 1978*). This may be due to metabolic competition amongst gonococcal strains, causing the population structure to segregate into distinct metabolic types (*Gupta, 2024*). Previous research has posited that selection can preserve such lineages even in the face of extensive HGT, as has been the case in immune selection for differing antigenic types in other bacterial species (*Gupta, 2024*; *Watkins et al., 2016*; *Buckee et al., 2011*). Sequence-based approaches including 7-locus MLST, NG-STAR, and cgMLST have confirmed that

the *N. gonorrhoeae* population contains identifiable groups of related isolates, some of which are associated with clinically relevant phenotypes such as AMR (*Harrison et al., 2020*; *Golparian et al., 2021*; *Golparian et al., 2024*; *Sánchez-Busó et al., 2022*), with outbreak events (*Smolarchuk et al., 2018*), or with specific at-risk groups such as men who have sex with men (MSM; *Sánchez-Busó et al., 2019*; *Williamson et al., 2019*). However, some of these typing systems are prone to providing misleading classifications, where isolates that are not closely related appear to be, due to the effects of HGT within the small numbers of loci used to classify them. Consequently, while equivalent systems have proved highly effective at categorising the closely related *N. meningitidis* (*Maiden, 2008*), the extensive HGT observed across the gonococcal genome necessitates a whole genome approach to reliably identify related groups of *N. gonorrhoeae* isolates (*Harrison et al., 2020*). LIN code achieves this by applying cgMLST, using hundreds of loci belonging to a wide range of functional categories, including genes involved in metabolic pathways, genetic processing, and AMR. This allows the LIN code to capture gonococcal population structure with increased consistency and higher resolution than many previous approaches, providing fresh insight into the biology that underlies this species' genetic composition.

The LIN nomenclature conveys lineage information in the form of hierarchical clustering at sequential thresholds of tolerated allelic mismatch within a cgMLST scheme (i.e. the number of alleles differing out of the total list of loci in the scheme; *Hennart et al., 2022*; *van Rensburg et al., 2024*). Here, 11 thresholds were used, resulting in an 11-bin barcode. The leftmost numbers of the barcode represent clustering at the highest thresholds of allelic mismatch, here corresponding with LIN superlineage (up to 550 mismatches), LIN lineage (300 mismatches), and LIN sublineage (125 mismatches). Progressing across the barcode to the right, each number indicates clustering at an increasing level of allelic similarity, ultimately resulting in the delineation of highly related isolates that differ in a very small number of core loci (*Figure 4*; *Palma et al., 2024*; *Hennart et al., 2022*; *van Rensburg et al., 2024*). Subsets of thresholds, for example a prefix of the first three or four, may be used to define clustering down to a particular level of taxonomic similarity (*Hennart et al., 2022*; *van Rensburg et al., 2024*). If two isolates are highly related and therefore identical across their core gene profile, they will share the same LIN code.

The use of multiple thresholds enables the LIN code to relate back to several familiar nomenclatures, such as core genome groups and 7-locus MLST, which are both broadly equivalent to LIN lineages and NG-STAR CCs, approximately equivalent to LIN sublineages. The difference lies in the accuracy and stability of the LIN code versus its predecessors. Visualisation via phylogenetic trees demonstrated that the LIN codes related strongly to phylogenetic structure, capturing distinct clades without overlap. This is in contrast to previous nomenclatures such as 7-locus MLST, which can form polyphyletic groups due to the misleading effects of HGT (*Harrison et al., 2020*).

Furthermore, the multi-resolution, hierarchical nature of LIN codes also allows this nomenclature to encode more information about the relationship between isolates than conventional typing systems (*Palma et al., 2024*). For example, if a pair of isolates belongs to MLST 1 and 2, it can only be deduced that they differ in between 1 and 7 housekeeping genes. Meanwhile, if the same pair of isolates belongs to LIN lineages 0_2_1 and 0_2_2, this communicates that these isolates differ by no more than 550/1430 core gene alleles, but by more than 300/1430, quickly defining the degree of their relatedness (*Figure 4*). This is useful when comparing isolates, for example those associated with AMR. Previous approaches used to track AMR strains include both 7-locus MLST and NG-STAR, which types isolates based on their alleles across seven AMR genes (*penA*, *mtrR*, *porB*, *ponA*, *gyrA*, *parC*, and 23 S rRNA; *Demczuk et al., 2017*). This typing system has been successfully used in many analyses of AMR associated gonococci, and NG-STAR CCs have been recommended as a tool for characterising gonococcal lineages (*Golparian et al., 2021*; *Golparian et al., 2024*). However, due to HGT, specific NG-STAR types or clonal complexes do not always represent closely related isolates. LIN sublineages were able to discern isolates with a similar level of resolution to NG-STAR CCs, but higher reliability. This will facilitate more accurate tracking of AMR associated isolates, such as those belonging to LIN sublineage 0_2_0_1, when compared to the equivalent NG-STAR CC 90, which also includes unrelated isolates belonging to LIN superlineages 0_0 and 1_1. While NG-STAR effectively characterises an isolate's AMR genotype (*Demczuk et al., 2017*), it cannot consistently distinguish related isolates. LIN code fulfils this role, while also providing increased information about the degree of relatedness of the isolates in question.

Accurately defining this lineage structure will be important to elucidate how AMR emerges in distinct gonococcal lineages.

As an allele-based method, LIN codes enable analysis of a large number of isolates rapidly and reproducibly (*Palma et al., 2024*; *Hennart et al., 2022*; *Vinatzer et al., 2017*). This is an improvement on phylogenetic tools used to identify gonococcal lineages which rely on nucleotide sequence alignments, requiring significant time and expertise to compute (*Tonkin-Hill et al., 2019*). LIN codes can replicate similar clustering, identifying the same lineages while being fully automated (*Palma et al., 2024*). For example, hierarchical Bayesian analyses (BAPs) have previously been used to delineate gonococcal population structure into two lineages: lineage A, associated with high-risk sexual networks and AMR, and lineage B, associated with lower risk networks and susceptibility (*Sánchez-Busó et al., 2019*). In our LIN code analysis, we observed that lineage A (BAPS 8, 7, and 9) corresponds to LIN lineages 0_2_0, 0_2_1 and 0_2_17, reaffirming that cgMLST is able to provide the same resolution as SNP-based techniques (*Harrison et al., 2020*). While BAPs analysis can be complex and time intensive to run, LIN codes can be used to rapidly identify lineage A gonococci simply by uploading WGS data to PubMLST. Within 24 hr of upload, any sequence data in which 1405/1430 core loci can be annotated (including private isolate records) will be assigned a Ng cgMLST v2 cgST; if this is a previously identified cgST, a LIN code will be assigned immediately; if a new cgST, the new LIN code will be assigned by the following Monday. Furthermore, these isolates can then be analysed in the context of the wider PubMLST gonococcal genome collection, which currently includes >28,000 public records and integrates accessible plugins such as GrapeTree (*Zhou et al., 2018*) and Genome Comparator (*Jolley et al., 2018*). This facilitates further investigation, for example in identifying an isolates' relationship to known AMR strains, or in the analysis of suspected transmission linked isolates from within the same community.

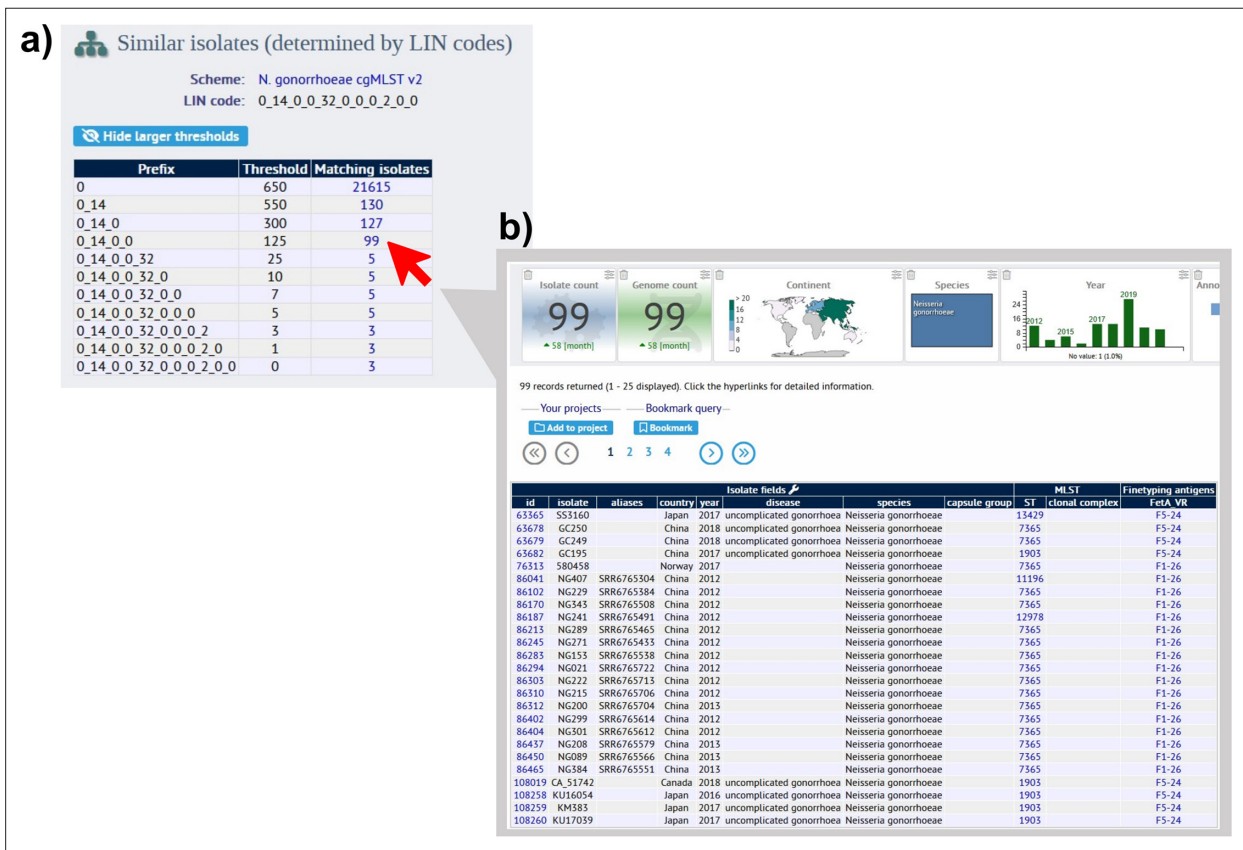

**Figure 8.** Using PubMLST to explore related isolates by LIN code. Within an isolate's information page, here using isolate SC18-25 (PubMLST id: 165303; https://pubmlst.org/bigsdb?page=info&db=pubmlst_neisseria_isolates&id=165303), it is possible to view a breakdown table of similar isolates by LIN code. This isolate shares a complete LIN code with 2 other isolates, meaning they are identical in their core genome. (**a**). Clicking on the 'matching isolates' number at a certain LIN code threshold then takes the user to the dataset of matching isolates for further analysis (**b**). This feature can be applied in the investigation of transmission chains, outbreak events and the dissemination of AMR through clonal expansion.

To further illustrate the efficacy of LIN code, we reproduced an analysis of 170 global ceftriaxone-resistant isolates (*Fifer et al., 2015*). The original article's methodology involved a WGS alignment and generation of a maximum likelihood tree in order to characterise eight major phylogroups (*Fifer et al., 2015*). The gonococcal LIN code instantly reproduced these clusters, while providing additional detail about each clade in the form of superlineage/sublineage divisions and simultaneously contextualising the isolates amongst the wider PubMLST database. For example, lineage 0_14_0 was over-represented in this ceftriaxone-resistant dataset when compared against 1000 randomly selected *N. gonorrhoeae* isolates from PubMLST. Also, LIN code was able to classify several divergent isolates that were not assigned to any phylogroup in the original publication and detected 15 instances of matching full-length LIN codes, meaning these isolates were identical across their core genome and could therefore represent isolates associated with transmission events (*Harrison et al., 2020*). The PubMLST interface facilitates in-depth analysis of these instances of shared LIN codes, allowing users to explore related isolates at various thresholds of allelic dissimilarity by viewing a table of similar isolates (as defined by LIN code) on an isolate's information page (*Figure 8*). This enables quick investigation of any related isolates, including their location, year, allele at a particular locus or classification by other typing schemes such as NG-STAR.

The genetic mix-and-matching performed by *N. gonorrhoeae* can make characterisation of its population structure difficult (*Harrison et al., 2020*). However, the LIN code nomenclature proposed here provides clarity, consistency, and stability in its description of *N. gonorrhoeae* lineages. The multi-resolution clustering intrinsic to LIN code facilitates a common language around lineage nomenclature at different epidemiological levels, from high divisions such as superlineage down to unique clones (*Palma et al., 2024*). In conclusion, *N. gonorrhoeae* LIN codes represent a portable, publicly available taxonomic nomenclature that has the potential to enhance surveillance of *N. gonorrhoeae* in order to benefit public health.

## Acknowledgements

The authors thank Nazreen Hadjirin and Iman Yassine for their valuable advice. Computational aspects of this work were enabled by the Oxford University Biomedical Research Computing (BMRC) facility.

## Additional information

### Funding

| Funder | Grant reference number | Author |
|---|---|---|
| Biotechnology and Biological Sciences Research Council | BB/M011224/1 | Anastasia Unitt |
| Wellcome Trust | 10.35802/218205 | Keith A Jolley<br>Martin CJ Maiden |
| Wellcome Trust | 10.35802/214374 | Kasia M Parfitt<br>Odile B Harrison |
| Ministry of Education, Indonesia | | Made A Krisna |
| Nuffield Department of Population Health, University of Oxford | | Anastasia Unitt<br>Odile B Harrison |

The funders had no role in study design, data collection and interpretation, or the decision to submit the work for publication. For the purpose of Open Access, the authors have applied a CC BY public copyright license to any Author Accepted Manuscript version arising from this submission.

### Author contributions

Anastasia Unitt, Conceptualization, Data curation, Formal analysis, Validation, Investigation, Visualization, Methodology, Writing – original draft, Writing – review and editing; Made A Krisna, Software,

Methodology; Kasia M Parfitt, Formal analysis; Keith A Jolley, Resources, Software, Validation, Methodology; Martin CJ Maiden, Conceptualization, Supervision, Project administration, Writing – review and editing; Odile B Harrison, Conceptualization, Resources, Supervision, Project administration, Writing – review and editing

### Author ORCIDs
Anastasia Unitt ![ORCID] https://orcid.org/0000-0003-1392-9786
Keith A Jolley ![ORCID] https://orcid.org/0000-0002-0751-0287
Odile B Harrison ![ORCID] https://orcid.org/0000-0002-1623-0295

Reviewer #3 (Public review): https://doi.org/10.7554/eLife.107758.3.sa1
Author response https://doi.org/10.7554/eLife.107758.3.sa2

## Additional files

### Supplementary files
Supplementary file 1. Table of dataset 1. This dataset comprises 896 *N. gonorrhoeae* isolates, collating a representative range of Ng cgMLST v1 core genome groups (at allelic mismatch threshold 300).

Supplementary file 2. Table of dataset 2. This dataset comprises 3935 *N. gonorrhoeae* isolates, collating a representative range of Ng cgMLST v1 core genome groups (at allelic mismatch threshold 300).

Supplementary file 3. Table of core gene functions. Summarising gene functions for 1430 core loci in Ng cgMLST v2, as described in PubMLST.

Supplementary file 4. Table summarising Ng cgMLST v1 VS Ng cgMLST v2 genes. This table describes the loci excluded from v2 present in v1, and the loci added to v2 that were not included in v1.

Supplementary file 5. Tables showing examples of MLST, NG-STAR CC, and NG-MAST-ST association with LIN lineage, LIN sublineage, and LIN code group.

Supplementary file 6. Table describing LIN lineages of isolates carrying Mosaic type 34 *penA* alleles. This table demonstrates that type 34 *penA* variants are associated with LIN lineage 266.

Supplementary file 7. Table of 170 isolates re-analysed from *Fifer et al., 2015*. Eight principal LIN code lineages were identified in this re-analysis.

MDAR checklist

### Data availability
All whole genome sequence data used in this publication is publicly available in the https://pubmlst.org/organisms/neisseria-spp database. The specific datasets used are described in the supplementary files. Computational scripts are documented at https://doi.org/10.17504/protocols.io.4r3l21beqg1y/v1 (*Unitt, 2025*).

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
