## [Editor Report · eLife Assessment]

This **important** study introduces the Life Identification Number (LIN) coding system as a powerful and versatile approach for classifying Neisseria gonorrhoeae lineages. The authors show that LIN codes capture both previously defined lineages and their relationships in a way that aligns with the species' phylogenetic structure. The **compelling** evidence presented, together with its integration into the PubMLST platform, underscores its strong potential to enhance epidemiological surveillance and advance our understanding of gonococcal population biology.

---

## [Referee Report · Reviewer #3 (Public review)]

Summary:

In this well-written manuscript, Unitt and colleagues propose a new, hierarchical nomenclature system for the pathogen Neisseria gonorrhoeae. The proposed nomenclature addresses a longstanding problem in N. gonorrhoeae genomics, namely that the highly recombinant population complicates typing schemes based on only a few loci and that previous typing systems, even those based on the core genome, group strains at only one level of genomic divergence without a system for clustering sequence types together. In this work, the authors have revised the core genome MLST scheme for N. gonorrhoeae and devised life identification numbers (LIN) codes to describe the N. gonorrhoeae population structure.

Strengths:

The LIN codes proposed in this manuscript are congruent with previous typing methods for Neisseria gonorrhoeae like cgMLST groups, Ng-STAR, and NG-MAST. Importantly, they improve upon many of these methods as the LIN codes are also congruent with the phylogeny and represent monophyletic lineages/sublineages. Additionally, LIN code cluster assignment is fixed, and clusters are not fused as is common in other typing schemes.

The LIN code assignment has been implemented in PubMLST allowing other researchers to assign LIN codes to new assemblies and put genomes of interest in context with global datasets, including in private datasets.

Weaknesses:

The authors have defined higher resolution thresholds for the LIN code scheme. However, they do not investigate how these levels correspond to previously identified transmission clusters from genomic epidemiology studies. This will be an important focus of future work, but it may be beyond the scope of the current manuscript.

Comments on revisions:

The authors have addressed my previous comments. I have no additional recommendations.

---

## [Author Response]

The following is the authors’ response to the original reviews.

**Reviewer #1 (Public review):**
Summary:Bacterial species that frequently undergo horizontal gene transfer events tend to have genomes that approach linkage equilibrium, making it challenging to analyze population structure and establish the relationships between isolates. To overcome this problem, researchers have established several effective schemes for analyzing N. gonorrhoeae isolates, including MLST and NG-STAR. This report shows that Life Identification Number (LIN) Codes provide for a robust and improved discrimination between different N. gonorrhoeae isolates.Strengths:The description of the system is clear, the analysis is convincing, and the comparisons to other methods show the improvements offered by LIN Codes.Weaknesses:No major weaknesses were identified by this reviewer.

We thank the reviewer for their assessment of our paper.

**Reviewer #2 (Public review):**
Summary:This paper describes a new approach for analyzing genome sequences.Strengths:The work was performed with great rigor and provides much greater insights than earlier classification systems.Weaknesses:A minor weakness is that the clinical application of LIN coding could be articulated in a more in-depth way. The LIN coding system is very impressive and is certainly superior to other protocols. My recommendation, although not necessary for this paper, is that the authors expand their analysis to noncoding sequences, especially those upstream of open reading frames. In this respect, important cis-acting regulatory mutations that might help to further distinguish strains could be identified.

We thank the reviewer for their comments. LIN code could be applied clinically, for example in the analysis of antibiotic resistant isolates, or to investigate outbreaks associated with a particular lineage. We have updated the text to note this, starting at line 432.

In regards to non-coding sequences: unfortunately, intergenic regions are generally unsuitable for use in typing systems as (i) they are subject to phase variation, which can occlude relationships based on descent; (ii) they are inherently difficult to assemble and therefore can introduce variation due to the sequencing procedure rather than biology. For the type of variant typing that LIN code represents, which aims to replicate phylogenetic clustering, protein encoding sequences are the best choice for convenience, stability, and accuracy. This is not to say that it is not a valid object to base a nomenclature on intergenic regions, which might be especially suitable for predicting some phenotypic characters, but this will still be subject to problem (ii), depending on the sequencing technology used. Such a nomenclature system should stand beside, rather than be combined with or used in place of, phylogenetic typing. However, we could certainly investigate the relationship between an isolates LIN code and regulatory mutations in the future.

**Reviewer #3 (Public review):**
Summary:In this well-written manuscript, Unitt and colleagues propose a new, hierarchical nomenclature system for the pathogen Neisseria gonorrhoeae. The proposed nomenclature addresses a longstanding problem in N. gonorrhoeae genomics, namely that the highly recombinant population complicates typing schemes based on only a few loci and that previous typing systems, even those based on the core genome, group strains at only one level of genomic divergence without a system for clustering sequence types together. In this work, the authors have revised the core genome MLST scheme for N. gonorrhoeae and devised life identification numbers (LIN) codes to describe the N. gonorrhoeae population structure.Strengths:The LIN codes proposed in this manuscript are congruent with previous typing methods for Neisseria gonorrhea, like cgMLST groups, Ng-STAR, and NG-MAST. Importantly, they improve upon many of these methods as the LIN codes are also congruent with the phylogeny and represent monophyletic lineages/sublineages.The LIN code assignment has been implemented in PubMLST, allowing other researchers to assign LIN codes to new assemblies and put genomes of interest in context with global datasets.Weaknesses:The authors correctly highlight that cgMLST-based clusters can be fused due n to "intermediate isolates" generated through processes like horizontal gene transfer. However, the LIN codes proposed here are also based on single linkage clustering of cgMLST at multiple levels. It is unclear if future recombination or sequencing of previously unsampled diversity within N. gonorrhoeae merges together higher-level clusters, and if so, how this will impact the stability of the nomenclature.The authors have defined higher resolution thresholds for the LIN code scheme. However, they do not investigate how these levels correspond to previously identified transmission clusters from genomic epidemiology studies. It would be useful for future users of the scheme to know the relevant LIN code thresholds for these investigations.

We thank the reviewer for their insightful comments. LIN codes do use multi-level single linkage clustering to define the cluster number of isolates. However, unlike previous applications of simple single linkage clustering such as N. gonorrhoeae core genome groups (Harrison et al., 2020), once assigned in LIN code, these cluster numbers are fixed within an unchanging barcode assigned to each isolate. Therefore, the nomenclature is stable, as the addition of new isolates cannot change previously established LIN codes.

Cluster stability was considered during the selection of allelic mismatch thresholds. By choosing thresholds based on natural breaks in population structure (Figure 3), applying clustering statistics such as the silhouette score, and by assessing where cluster stability has been maintained within the previous core genome groups nomenclature, we can have confidence that the thresholds which we have selected will form stable clusters. For example, with core genome groups there has been significant group fusion with clusters formed at a threshold of 400 allelic differences, while clustering at a threshold of 300 allelic differences has remained cohesive over time (supported by a high silhouette score) and so was selected as an important threshold in the gonococcal LIN code. LIN codes have now been applied to >27000 isolates in PubMLST, and the nomenclature has remained effective despite the continual addition of new isolates to this collection. The manuscript emphasises these points at line 96 and 346.

Work is in progress to explore what LIN code thresholds are generally associated with transmission chains. These will likely be the last 7 thresholds (25, 10, 7, 5, 3, 1, and 0 allelic differences), as previous work has suggested that isolates linked by transmission within one year are associated with <14 single nucleotide polymorphism differences (De Silva et al., 2016). The results of this analysis will be described in a future article, currently in preparation.

Harrison, O.B., et al. Neisseria gonorrhoeae Population Genomics: Use of the Gonococcal Core Genome to Improve Surveillance of Antimicrobial Resistance. The Journal of Infectious Diseases 2020.

De Silva, D., et al. Whole-genome sequencing to determine transmission of Neisseria gonorrhoeae: an observational study. The Lancet Infectious Diseases 2016;16(11):1295-1303.

**Reviewer #3 (Recommendations for the authors):**
(1) Data/code availability: While the genomic data and LIN codes are available in PubMLST and new isolates uploaded to PubMLST can be assigned a LIN code, it is also important to have software version numbers reported in the methods section and code/commands associated with the analysis in this manuscript (e.g. generation of core genome, statistical analysis, comparison with other typing methods) documented in a repository like GitHub.

Software version numbers have been added to the manuscript. Scripts used to run the software have been compiled and documented on protocols.io, DOI: dx.doi.org/10.17504/protocols.io.4r3l21beqg1y/v1

(2) Line 37: Missing "a" before "multi-drug resistant pathogen".

This has been corrected in the text.

(3) Line 60: Typo in geoBURST.

The text refers to a tool called goeBURST (global optimal eBURST) as described in Francisco, A.P. et al., 2009. DOI: 10.1186/1471-2105-10-152. Therefore, “geoBURST” would be incorrect.

(4) Line 136-138: It might be helpful to discuss how premature stop codons are treated in this scheme. Often in isolates with alleles containing early premature stop codons, annotation software like prokka will annotate two separate ORFs, which are then clustered with pangenome software like PIRATE. How does the cgMLST scheme proposed here treat premature stop codons? Are sequences truncated at the first stop codon, or is the nucleotide sequence for the entire gene used even if it is out of frame?

In PubMLST, alleles with premature stop codons are flagged, but otherwise annotated from the typical start to the usual stop codon, if still present. This also applies to frameshift mutations – a new unique allele will be annotated, but flagged as frameshift. In both cases, each new allele with a premature stop codon or frameshift will require human curator involvement to be assigned, to ensure rigorous allele assignment. As the Ng cgMLST v2 scheme prioritised readily auto-annotated genes, loci which are prone to internal stop codons or frameshifts with inconsistent start/end codons are excluded from the scheme. The text has been updated at line 128 to mention this.

(5) Line 213-214: What were the versions of software and parameters used for phylogenetic tree construction?

Version numbers have been added to the text between lines 214-219. Parameters have been included with the scripts documented at protocols.io DOI: dx.doi.org/10.17504/protocols.io.4r3l21beqg1y/v1

(6) Line 249: K. pneumoniae may also be a more diverse/older species than N. gonorrhoeae.

The text has been updated at line 252-253 to emphasize the difference in diversity. The age of N. gonorrhoeae as a species is a matter of scientific debate, and out of the scope of this paper to discuss.

(7) Line 278-279: Were some isolates unable to be typed, or have they just been added since the LIN code assignment occurred?

Some genomes cannot be assigned a LIN code due to poor genome quality. A minimum of 1405/1430 core genes must have an allele designated for a LIN code to be assigned. Genomes with large numbers of contigs may not meet this requirement. LIN code assignment is an ongoing process that occurs on a weekly basis in PubMLST, performed in batches starting at 23:00 (UK local time) on Sundays. The text has been updated to describe this at lines 196 and 282-283.

(8) Line 314-315: Was BAPS rerun on the dataset used in this manuscript, or is this based on previously assigned BAPS groups?

This was based on previously assigned BAPs groups, as described between lines 315-320.

(9) Line 421-423: Are there options for assigning LIN codes that do not require uploading genomes to PubMLST? I can imagine that there may be situations where researchers or public health institutions cannot share genomic data prior to publication.

Isolate data does not need to be shared to be uploaded and assigned a LIN code in PubMLST. data owners can create a private dataset within PubMLST viewable only to them, on which automated assignment will be performed. LIN code requires a central repository of genomes for new codes to be assigned in relation to. The text has been updated to emphasize this at line 197 and 427.

(10) Figure 6: How is this tree rooted? Additionally, do isolates that have unannotated LIN codes represent uncommon LIN codes or were those isolates not typed?

The tree has been left unrooted, as it is being used to visualise the relationships between the isolates rather than to explore ancestry. Detail on what LIN codes have been annotated can be found in the figure legend, which describes that the 21 most common LIN code lineages in this 1000 isolate dataset have been labelled. All 1000 isolates used in the tree had a LIN code assigned, but to ensure good legibility not all lineages were annotated on the tree. The legend has been updated to improve clarity.